# In Vitro Probiotic Properties and DNA Protection Activity of Yeast and Lactic Acid Bacteria Isolated from A Honey-Based Kefir Beverage

**DOI:** 10.3390/foods8100485

**Published:** 2019-10-12

**Authors:** Bruna de Oliveira Coelho, Fernanda Fiorda-Mello, Gilberto V. de Melo Pereira, Vanete Thomaz-Soccol, Sudip K. Rakshit, Júlio Cesar de Carvalho, Carlos Ricardo Soccol

**Affiliations:** 1Department of Bioprocess Engineering and Biotechnology, Federal University of Paraná (UFPR), 81531-980 Curitiba, PR, Brazil; bdeoliveiracoelho@gmail.com (B.d.O.C.); gilbertovinicius@gmail.com (G.V.d.M.P.); vanetesoccol@gmail.com (V.T.-S.);; 2Department of Food Engineering, Federal University of Paraná (UFPR), 81531-980 Curitiba, PR, Brazil; fernandafiorda@gmail.com; 3Department of Chemical Engineering, Lakehead University, Thunder Bay, ON P7B 5E1, Canada; skrait57@gmail.com

**Keywords:** non-dairy products, functional beverage, kefir fermentation, DNA protection, comet assay, *Saccharomyes cerevisiae*

## Abstract

The probiotic characteristics of three acid-tolerant microbial strains, *viz.,*
*Lactobacillus satsumensis* LPBF1, *Leuconostoc mesenteroides* LPBF2 and *Saccharomyes cerevisiae* LPBF3, isolated from a honey-based kefir functional beverage, were studied following the requirements established by the Food and Agriculture Organization of the United Nation/World Health Organization (FAO/WHO), including host-associated stress resistance, epithelium adhesion ability, and antimicrobial activity. The three microbial strains tolerated different pH values (2.0, 3.0, 4.0 and 7.0) and bile salt concentrations (0.3% and 0.6%), and survive in the presence of simulated gastric juice, which are conditions imposed by the gastrointestinal tract. In addition, they showed high percentages of hydrophobicity, auto aggregation and anti-pathogenic against *Escherichia coli* and *Staphylococcus aureus*, with no hemolytic activity. The protective capacity of human DNA through microbial treatment was investigated by single-cell gel electrophoresis (SCGE) comet assay. The three selected strains showed DNA protection effect against damage caused by hydroxyl radical (H_2_O_2_). However, when the *S. cerevisiae* treatment was applied, the most effective DNA protection index was observed, which can be associated to its high production of extracellular antioxidants as reveled by the 2,2-diphenyl-1-picryl-hydrazylhydrate (DPPH) method. These results indicated that the three selected microbial strains could be useful for preventing oxidative DNA damage and cellular oxidation in food products. As well-adapted microbial cells, the selected strains can be used for production of non-dairy functional beverages, especially for vegans and/or vegetarians and lactose intolerants.

## 1. Introduction

Probiotics are defined as viable microorganisms, which upon ingestion in an appropriate concentration exert health benefits on the host [1]. Promising probiotic microorganisms include members of the genera *Lactobacillus*, *Bifidobacterium*, *Leuconostoc* and *Saccharomyces* [2,3,4,5]. Kefir—a beverage commonly manufactured by fermenting milk with kefir grains—is known to be an excellent source of probiotic microorganisms. The complex microbial symbiosis contained in kefir grains results in a naturally carbonated beverage (associated with yeast metabolism) with an acidic taste and creamy consistency due to lactic acid bacteria (LAB) metabolism. The consumption of kefir beverage has been associated with beneficial effects on human health, and several LAB and yeasts found in kefir have been recognized as probiotics [6,7,8].

The Food and Agriculture Organization of the United Nations/World Health Organization (FAO/WHO) [9] published the “Guidelines for Evaluation of Probiotics in Food”, which established safety and effectiveness standards for probiotics. In this guideline, several criteria are suggested for the selection of probiotics, including resistance to unfavorable conditions that the human body imposes, epithelium adhesion ability, antimicrobial activity and safety assessment. Recently, we have proposed extending these methods to perform tests proving that selected probiotics have functional proprieties and a safety status [10]. These include the secretion of functional molecules and DNA protection activity, detection of antibiotic resistance and harmful metabolites production.

A growing number of scientific studies have involved the selection of new strains with different and specific functional properties. The isolation and application of probiotic microorganisms from non-dairy sources can extend the possibility of recovering strains with improved functional characteristics [11,12]. Honey-based kefir is a newly developed functional beverage with high antioxidant activity, exopolysaccharides content and DNA protection effect, containing a diverse number of potential probiotic microorganisms, including *Lactobacillus satsumensis*, *Leuconostoc mesenteroides*, *Bacillus megaterium* and *Saccharomyces cerevisiae* [13]. Here we study the probiotic potential of microorganisms (yeast and lactic acid bacteria) isolated from honey kefir fermentation following the requirements established by FAO/WHO, including the extending methods proposed by our group such as potential antioxidant activity and DNA protection ability.

## 2. Materials and Methods

### 2.1. Microorganisms

A total of 75 strains (39 bacteria and 36 yeasts), isolated from honey kefir beverage [13] were used in this study. Among these, *Lactobacillus satsumensis* LPBF1, *Leuconostoc mesenteroides* LPBF2 and *Saccharomyces cerevisiae* LPBF3 were pre-selected based on their ability to tolerate the effects of low pH. The identification of these three potential probiotic strains was confirmed by the 16S rRNA gene and Internal Transcribed Spacer (ITS) region sequencing, for bacteria and yeast, respectively [14]. The strains were maintained as frozen (−80 °C) stock cultures in De Man, Rogosa and Sharpe (MRS) broth (for bacteria) and Yeast Malt (YM) broth (for yeast) containing 20% (*v*/*v*) glycerol.

### 2.2. Stress Tolerance

#### 2.2.1. Acid Tolerance

The resistance under acid conditions was carried out according to Pieniz et al. [15], with some modifications. Bacteria and yeast cells were grown in MRS broth at 37 °C and YM broth at 30 °C, respectively, for 24 h. The cultures were standardized at an optical density (OD600) = 1.0 ± 0.05. One milliliter of standardized culture was added into tubes containing 9 mL of respective sterile broth (MRS or YM) with the following pH values: 2.0, 3.0, 4.0 (adjusted with HCl) and 7.0 (adjusted with NaOH). Viable cell counts were determined after exposure to acidic conditions for 0, 1, 2, 3 and 4 h. Survival cell counts were expressed as log values of colony-forming units per mL (CFU/mL) by the pour plate method after serial dilutions in 0.1% saline-peptone water. The survival percentage was calculated as the following equation:% Survival = final (CFU/mL)/initial (CFU/mL) × 100.

#### 2.2.2. Resistance to Bile Salts

After strains were grown in the MRS broth (for bacteria) and YM broth (for yeast), cells were harvested by centrifugation (10,000× *g* for 10 min at 4 °C), washed three times with 0.1 M phosphate buffered saline (PBS) (pH 7.2) and suspended in a 0.5% NaCl solution. The cultures were standardized at an optical density (OD600) = 1.0 ± 0.05. A 0.2 mL aliquot of suspensions were inoculated into 1.0 mL of YM broth (yeast) and MRS broth (LAB) with 0% (control-pH 7.0), 0.3 and 0.6% (*w*/*v*) of bile salts, at pH 7.4. Total viable counts were determined after exposure to bile salts solution at 0, 1, 2, 3 and 4 h of incubation by pour plate method after serial dilutions and incubated at 37 °C (for bacteria) or 30 °C (for yeast) for 24 h. Values were expressed as log CFU/mL [16].

#### 2.2.3. Survival in Simulated Gastrointestinal Tract

Survival in simulated gastrointestinal tract was performed according to Pieniz et al. [15]. After 24 h of incubation in MRS broth at 37 °C (for bacteria) or YM broth at 30 °C (for yeast), cells were harvested by centrifugation (10,000× *g* for 10 min at 4 °C), washed three times with 0.1 M phosphate buffered saline (PBS) (pH 7.2) and suspended in 0.5% NaCl solution. The cultures were standardized at an optical density (OD600) = 1.0 ± 0.05. A 0.2 mL aliquot of suspensions were inoculated into 1.0 mL of simulated gastric or intestinal juices and incubated at 37 °C for 4 h. Survival cell counts were determined at initial time (0 h) and 1, 2, 3 and 4 h for the gastric tolerance and intestinal tolerance. Values were expressed as log CFU/mL. Simulated gastric juice was prepared fresh daily containing 3 mg of pepsin, 1 mL of NaCl solution (0.5%) and acidified with HCl to pH 3.0. Simulated intestinal juice was consisted of 1 mg of pancreatin, 1 mL of NaCl solution (0.5%) and adjusted to pH 8.0. Both solutions were sterilized by filtration through a 0.22 mm membrane.

### 2.3. Adhesion Ability

#### 2.3.1. Hydrophobicity

The test was conducted according to Chelliah et al. [17], with some modifications. A culture of 48 h of each strain was harvested by centrifugation (4000× *g* for 10 min at 4 °C). The pellets were washed twice with PBS and resuspended in the same buffer. The OD600 was adjusted to 0.6–0.8, and 5 mL of each suspension transferred to two tubes, containing 1 mL of xylene and 1 mL of toluene each. The tubes were agitated in a vortex and incubated at 37 °C. The absorbance of the solutions’ superior and inferior phase was measured with 30 and 60 min in a spectrophotometer at 600 nm. The hydrophobicity was determinate by the following formula:Hydrophobicity (%) = (Solvent layer absorbance − Aqueous layer absorbance)/Solvent layer absorbance

#### 2.3.2. Aggregation

Aggregation capacity was performed according to Ogunremi et al. [18]—with some modifications. Bacteria and yeast cells were grown in MRS broth at 37 °C and YM broth at 30 °C, respectively, for 48 h. The cultures were centrifuged at 3500× *g* for 5 min and resuspended with PBS buffer. The OD600 was adjusted to 1, and 4 mL of each suspension was transferred to round bottom tubes and agitated in a vortex. The absorbance was measured immediately after 5 and 24 h. Aggregation was determined according to the following formula:(1 − At/A0) × 100where At corresponds to the absorbance values obtained on different times points (t = 5 and 24 h); and A0 corresponds to the initial time absorbance (0 h). The suspensions were stained with methylene blue at 24 h and monitored by contrast microscopy at 100× magnification.

### 2.4. Anti-Pathogenic Activity

#### 2.4.1. Antimicrobial

The pathogens *Escherichia coli* JM109 and *Staphylococcus aureus* ATCC^®^ 6538, belonging to the collection of Biorefining Research Institute (Lakehead University, Thunder Bay, ON, Canada), were used in this study as photogenic microorganisms. Both pathogens were grown in a nutrient broth at 37 °C for 24 h and suspended in 0.85% NaCl solution standardized to OD600 of 0.150 in spectrophotometer, which corresponded to a 0.5 McFarland turbidity standard solution. One aliquot of 50 µL of culture containing grown *Lactobacillus satsumensis* LPBF1, *Leuconostoc mesenteroides* LPBF2 and *Saccharomyces cerevisiae* LPBF3 was applied onto wells on Mueller Hinton agar plates, previously inoculated with a swab soaked in a culture of each indicator bacteria. A 50 µL of honey kefir beverage was evaluated as well in this step to analyze whether antimicrobial activity would increase or decrease when the strains are in symbiosis. The plates were incubated at 37 °C and inhibition zones were measured after 24 h. Ampicillin (50 mg mL^−1^) was used as standard. The diameter of inhibition zones was measured using a caliper rule and halos ≥7 mm were considered inhibitory [15].

#### 2.4.2. Co-Aggregation

Probiotic and pathogenic cultures were prepared at the same conditions described in the aggregation assay [18]. A volume of 2 mL from *E. coli* and *S. aureus* suspensions were transferred to 2 mL of each probiotic strain tubes. The mixtures were agitated in a vortex and the absorbance was measured immediately after 5 and 24 h. Tubes containing only probiotic strains were used as negative controls. Samples were stained with methylene blue. Coaggregation was calculated according to the equation:Co-aggregation (%) = {[(Ax + Ay) /2 − A (x + y)/Ax + Ay/2]/[Ax + Ay/2]} × 100where A corresponds to absorbance; X and Y to each strain at negative control tubes; and X + Y to the mixture of probiotic and pathogenic strains.

### 2.5. Functional Characteristics

#### 2.5.1. Antioxidant Activity by the DPPH Method

The intracellular and extracellular antioxidant contents were measured according to Brand et al. [19], with some modifications. For extraction of intracellular antioxidants, 1 mL of each strain suspension was adjusted to Macfarland’s 0.5 scale, and the intracellular content was obtained by ultrasonic homogenizer for 5/1 min intervals (5 min on/1 min off, 35% amplitude), with constant cooling. The cell debris was removed by centrifugation at 5000× *g* for 10 min, and the supernatant was used for the antioxidant assay. First, 1 mL of the supernatant was added to 1 mL of the 2,2-diphenyl-1-picryl-hydrazylhydrate (DPPH) solution (0.15 mM in methanol). The mixture was incubated for 30 min, in the dark, and the absorbance was measured at 517 nm. The same procedure was performed to evaluate extracellular antioxidant content of each strain suspension. The control was methanol and DPPH solution, and the blank contained the suspension and methanol. The extracellular antioxidant content was measured according to formula “A” and intracellular content by the equation “B”:(A) Scavenging activity (%) = [1 − (Asample − Ablank)/Acontrol] × 100
(B) Scavenging activity (%) = [(Acontrol − Asample)/Acontrol] × 100where Asample corresponds to the absorbance of the sample; Ablank to the absorbance of the strain suspension and methanol; and Acontrol to the absorbance of methanol and DPPH.

#### 2.5.2. Single-Cell Gel Electrophoresis (SCGE) Assay (Comet Assay)

The comet assay proposed by Singh et al. [20] and improved by Collins [21] was used to evaluate DNA protection ability of the selected microbial strains. Microscope slides were covered with agarose one day before use for overnight dry. Subsequently, suspensions of 10^8^ cells of each microbial strain (*Lactobacillus satsumensis* LPBF1, *Leuconostoc mesenteroides* LPBF2 and *Saccharomyes cerevisiae* LPBF3) were prepared, respectively, and combined with lymphocytes separated from whole blood. The suspensions were exposed to hydrogen peroxide (30%) for 1 h and 24 h. As the negative control, a suspension containing only lymphocytes was added. For positive control, a suspension with lymphocytes and hydrogen peroxide was used. After exposure, agarose low melting point was added. The suspensions plus agarose were placed in slides. The slides were treated with lysis solution (1 mL Triton-X + 10 mL DMSO + 89 mL stock solution: 2.5 M NaCl; 100 mM EDTA; 10 mM Tris; 8 g NaOH; 1% Na lauroyl sarcosinate; pH 10) for 1 h at 8 °C. The slides were washed with PBS 1x and placed in an electrophoresis chamber (22 V and 300 mA for 20 min). Slides were stained with silver nitrate and dried at room temperature. For damage classification, cells with circular shape were considered as no damaged, and cells with “comet” shape were considered as DNA damage. The cells were classified in five categories, corresponding to the amount of damage: 0, no damage (<5%); 1, low level of damage (5–20%); 2, medium level of damage (20–40%); 3, high level of damage (40–95%); and total damage (>95%). The damage index (DI) was calculated as: DI = score/total of cells. Score was calculated with the following equation:Score = Damage 0 + Damage 1 + Damage 2 + Damage 3 + Damage 4where Damage 0 = 0 × n° of cells; Damage 1= 1 × n° of cells; Damage 2 = 2 × n° of cells; Damage 3 = 3 × n° of cells and Damage 4 = 4 × n° of cells.

### 2.6. Safety Assessment (Hemolytic Activity)

The selected microbial strains were tested for hemolytic activity using blood agar (7% *v*/*v* sheep blood) for 48 h incubation at 37 °C [22]. Strains that produced green-hued zones around the colonies (α-hemolysis) or did not produce any effect on the blood plates (γ-hemolysis) were considered non-hemolytic. Strains displaying blood lyses zones around the colonies were classified as hemolytic (β-hemolysis).

### 2.7. Statistical Analyses

The results were expressed as mean ± standard deviation from three replicate determinations. Differences were analyzed using one-way analysis of variance (ANOVA) followed by Tukey’s post-hoc test. *p*-values < 0.05 were considered to be statistically significant.

## 3. Results and Discussion

### 3.1. Stress Tolerance

A total of 39 LAB (including strains of *Leuconostoc mesenteroides*, *Lactobacillus satsumensis* and *Lysinibacillus sphaericus*) and 36 yeasts (including strains of *Hanseniaspora uvarum*, *Issatchenkia orientalis*, *Lachancea fermentati*, *Pichia membranifaciens*, *P. kudriavzevii*, *Saccharomyces cerevisiae* and *Zygosaccharomyces fermentati*), isolated from honey kefir beverage [13], were screened based on their ability to tolerate the effects of low pH (data not shown). Three acid resistant strains, *Lac. satsumensis* LPBF1, *Leuc. mesenteroides* LPBF2 and *S. cerevisiae* LPBF3, were pre-selected for further characterization.

After ingestion, probiotic cells must resist to antimicrobial factors in the stomach (low pH, gastric juice, and pepsin) and intestines (pancreatin and bile salts) [10]. These resistance properties were evaluated to ensure that the three candidate probiotics could withstand the stressful conditions of the human digestive system. Firstly, the microbial strains were analyzed in vitro for their ability to survive for some time under acidic conditions, in the presence of 0.3 and 0.6% bile salts and pancreatin, in a simulated intestinal juice (Figure 1 and Figure 2). The three strains survived in all times tested (1, 2, 3 and 4 h) at pH 2, pH 3, pH 4 and pH 7, maintaining high counts at pH 3 for 2 h, which are considered to be the standard values of acid tolerance of probiotic cultures [23]. All three selected strains also survived at 0.3 and 0.6% bile salts concentrations; the viable cells count remained above the limit of 10^3^ CFU mL^−1^ after 2 h, in which, according to Likotrafiti et al. [24], is the detection limit for probiotic strains resistance to bile salts. It has been hypothesized that this tolerance is associated with the ability of probiotics to reduce the detergent effect of bile salts by producing hydrolytic enzymes. Bile tolerance by probiotics has been revealed to be dependent on the bile type and strain, with resistance levels ranging from bile concentrations of 0.125–2.0% [25]. In general, bacteria cells are more tolerant than yeasts due to the capsule that surrounds the microbial cell. *S. cerevisiae* LPBF3 was more sensitive than bacteria strains to bile salts. Nevertheless, the yeast strain reached up to 10^4^ CFU mL^−1^ after 4 h of incubation, even at 0.6% of bile salts.

The ability of transit through the gastrointestinal tract was assessed by cultivating microbial strains in the presence of gastric juice containing pepsin or pancreatin (Figure 2). When exposed to both simulated gastric and intestinal conditions for 4 h, the strains analyzed exhibited cell count nearby 10^7^ CFU·mL^−1^. This allows the three probiotic strains to pass through the stomach.

### 3.2. Adhesion Ability

The microbial cell auto-aggregation ability ensures that the probiotic reaches a high cell density in the gut contributing to the adhesion mechanism, and cell surface hydrophobicity allows an improved interaction between microbe and human epithelial cells [10]. Auto-aggregation and hydrophobicity of the three selected microbial strains are showed in Figure 3. *Lac. satsumensis* LPBF1 and *S. cerevisiae* LPBF3 were stable at the time analyzed, reaching over 80% of aggregation after 24 h. On the other hand, *Lac. satsumensis* LPBF1 increased its aggregation over time, passing from 40% to 72% after 24 h. These auto aggregation values indicate that the probiotics analyzed can reach a high cell density in the gut contributing to the adhesion mechanism [26].

Hydrophobicity results for *Lac. satsumensis* LPBF1 and *S. cerevisiae* LPBF3 are showed in Figure 3, while no hydrophobicity ability was observed for *Leuc. mesenteroides* LPBF2. The hydrophobicity of *S. cerevisiae* LPB3 increased with time of exposure, while high values were observed for *Lac. satsumensis* LPBF1 at 30, 60 and 90 min of incubation (Figure 3). Cells that have toluene affinity are strong electron donors, with a good capacity of intestine colonization. Yeasts have a high affinity to organic solvents, such as toluene and xylene. *S. cerevisiae* LPBF3 was 67% hydrophobic in toluene and 78% in xylene within 60 min, which are similar values to those reported by Chelliah et al. [17] for *Pichia kudriavzevii*.

### 3.3. Anti-Pathogenic Assay

Once adhered to the gut, probiotics produce extracellular antimicrobial components through the conversion of carbohydrates, proteins and other minor compounds into important substances that can kill pathogenic bacteria. The pathogens included in this study (*E. coli* and *S. aureus*) are used as standards in antimicrobial activity tests [27,28,29,30,31]. The highest antimicrobial activities were observed for *Lac. satsumensis* LPBF1 against *E. coli* and *Leuc. mesenteroides* LPBF2 against *S. aureus* (Table 1). At this step, antimicrobial activities of honey kefir beverage were included against these same pathogens. Interestingly, the results showed higher antimicrobial activity when using the beverage containing the three microbial cultures. This demonstrates that the use of co-cultures of the three microbial strains can optimize the antimicrobial activity of the final product.

Co-aggregation is another antagonistic probiotic activity that provides pathogen agglomeration with probiotic cells and facilitates its elimination through feces. In this work, the co-aggregation of the three selected strains was tested with *E. coli* and *S. aureus* (Figure 4). The values for *Leuc. mesenteroides* LPBF2 and *S. cerevisiae* LPBF3 were similar for both pathogens, reaching 52% and 51% with *E. coli* and 2% and 6% with *S. aureus* after 24 h, respectively. *Saccharomyces cerevisiae* LPBF3 did not show any co-aggregation with *E. coli*, and a 22% of aggregation value was observed with *S. aureus*. These results corroborate with previous studies that evaluated yeast and lactobacilli co-aggregation with different pathogens [17,32,33].

### 3.4. Functional Characteristics

When passing through the GTI, probiotic cells secrete functional molecules which confer health benefits to the host. Among these, antioxidant compounds act against free radicals to resist their damaging effects to vital biomolecules and the tissues of the human body. The three selected microbial strains showed the ability to produce intra and extracellular antioxidants in different proportions, as reveled by the DPPH method (Figure 5). *L. satsumensis* and *S. cerevisiae* had the highest intracellular and extracellular activity, respectively. Thus, ingestion of these microbial strains could increase the amount of antioxidants in the body by extracellular secretion or by lysis of microbial cells (intracellular components) in the GTI.

The ability to protect human DNA through microbial treatment was investigated by comet assay (Figure 6). In lymphocyte cells treated with only H_2_O_2_, DNA was severely damaged (Figure 6E) in comparison to the negative control (Figure 6D). However, when microbial strains were applied, the amounts of damaged DNA were significantly reduced (Figure 6A–C); despite the occurrence of fragmented DNA tails in the images with the microorganism treatments, it is possible to observe the colloidal structure (or compaction) of the DNA. In Table 2, it was evidenced that microbial treatments reduced the DNA damage index compared to the positive control. Interestingly, *S. cerevisiae* treatment showed the best results after 1 and 24 h of H_2_O_2_ application, which could be associated to the higher production of extracellular antioxidants in relation to bacterial cells. Altogether, these results indicated that the three selected microbial strains could be useful for preventing oxidative DNA damage and cellular oxidation in pharmaceutical and food industries.

### 3.5. Safety Assessment

None of the microbial strains selected caused γ-, α- or β-hemolysis after 48 h incubation in blood agar plates (data not showed). The determination of hemolytic activity is considered a safety aspect for probiotic strains selection [9]. Yeast and lactobacilli isolated from foods matrixes has, in general, a very low frequency of hemolytic activity [34,35].

## 4. Conclusions

The microbial strains analyzed in this study, *Lac. satsumensis* LPBF1, *Leuc. mesenteroides* LPBF2, and *S. cerevisiae* LPBF3, fulfill the selection criteria established by the FAO/WHO for a candidate probiotic, including host-associated stress resistance, epithelium adhesion ability, and antimicrobial activity. They tolerated low pH, bile salts, and simulated gastric or intestinal juices, which are conditions imposed by the gastrointestinal tract. Further, they showed high percentages of hydrophobicity, auto-aggregation, and anti-pathogenic against *Escherichia coli* and *Staphylococcus aureus* (coaggregation and direct antimicrobial ability), with no hemolytic activity. In addition, these microbial strains were able to protect human DNA against damage caused by hydroxyl radical and showed high antioxidant activity. As species adapt and isolate from a non-dairy matrix, they can be used for the production of non-dairy functional beverage, especially for vegans and/or vegetarians and lactose intolerant consumers. In addition, the selection of the novel potential probiotic *S. cerevisiae* LPBF3 with its high DNA protection activity is an advance, since yeast is a well-recognized microbial group for its nonsusceptibility to antibiotics and good tolerance to industrial processing conditions (lyophilization and high temperatures).

## Figures and Tables

**Figure 1 foods-08-00485-f001:**
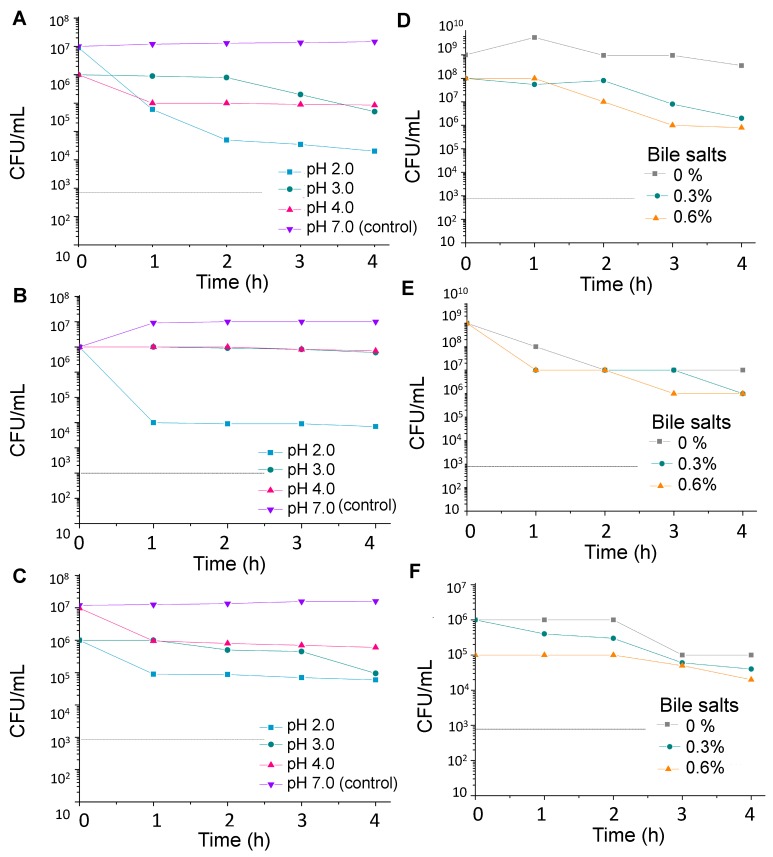
Stress tolerance of selected microbial strains isolated from a honey-based kefir beverage. Acid tolerance of *Lactobacillus satsumensis* LPBF1 (**A**), *Leuconostoc mesenteroides* LPBF2 (**B**) and *Saccharomyes cerevisiae* LPBF3 (**C**); bile salt tolerance of *Lactobacillus satsumensis* LPBF1 (**D**), *Leuconostoc mesenteroides* LPBF2 (**E**) and *Saccharomyes cerevisiae* LPBF3 (**F**). The dotted line is the detection limit.

**Figure 2 foods-08-00485-f002:**
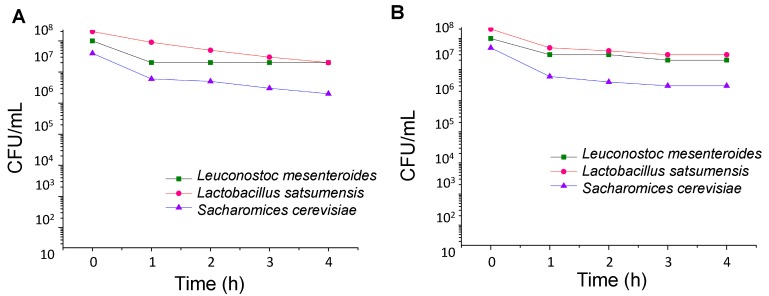
Resistance to simulated gastric juice containing pepsin (**A**) and intestinal juice containing pancreatin (**B**) of *Lactobacillus satsumensis* LPBF1, *Leuconostoc mesenteroides* LPBF2, and *Saccharomyes cerevisiae* LPBF3 kefir strains.

**Figure 3 foods-08-00485-f003:**
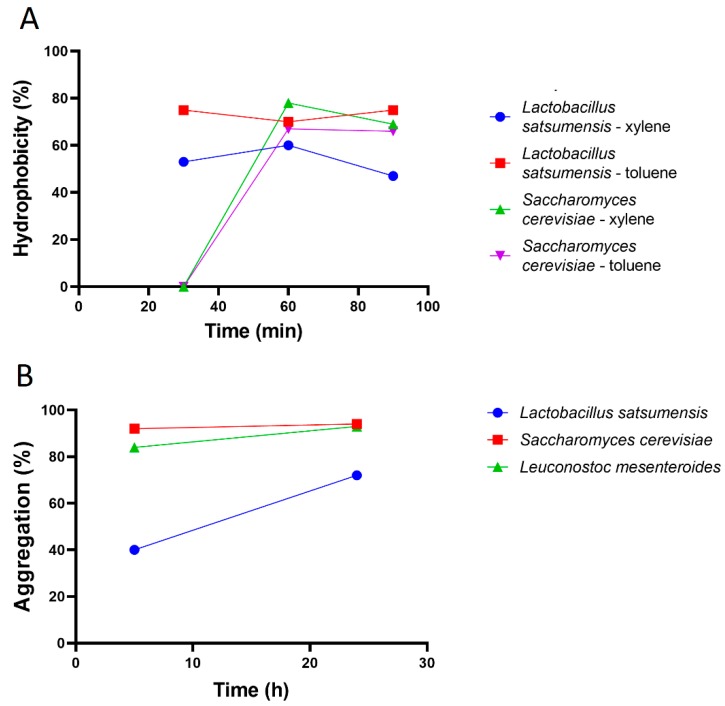
Hydrophobicity (**A**) of *Lactobacillus satsumensis* LPBF1 and *Saccharomyes cerevisiae* LPBF3 in 30, 60 and 90 min with xylene and toluene, and aggregation results (**B**) by 5 and 24 h. No hydrophobicity ability was observed for *Leuconostoc mesenteroides* LPBF2.

**Figure 4 foods-08-00485-f004:**
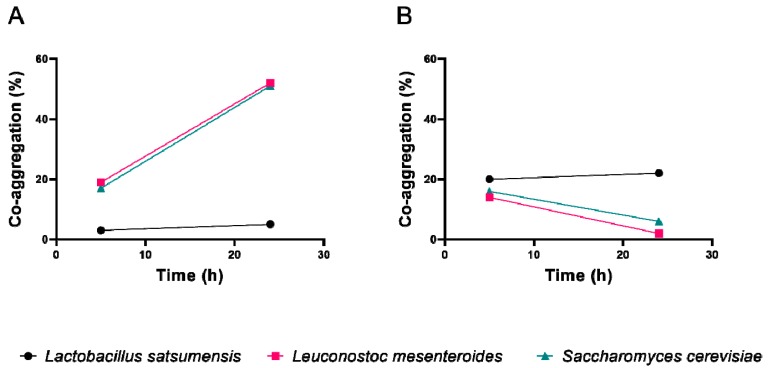
Co-aggregation of Lactobacillus satsumensis LPBF1, Leuconostoc mesenteroides LPBF2 and Saccharomyes cerevisiae LPBF3 with Escherichia coli (**A**) and Staphylococcus aureus (**B**) in 5 and 24 h.

**Figure 5 foods-08-00485-f005:**
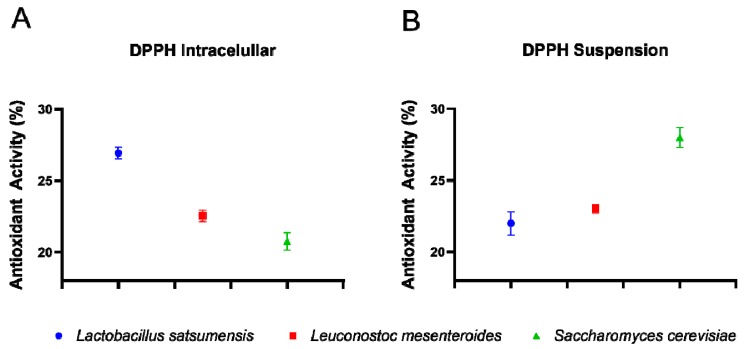
Antioxidant activity of intracellular (**A**) and extracellular (**B**) cells of *Lactobacillus satsumensis* LPBF1, *Leuconostoc mesenteroides* LPBF2 and *Saccharomyes cerevisiae* LPBF3.

**Figure 6 foods-08-00485-f006:**
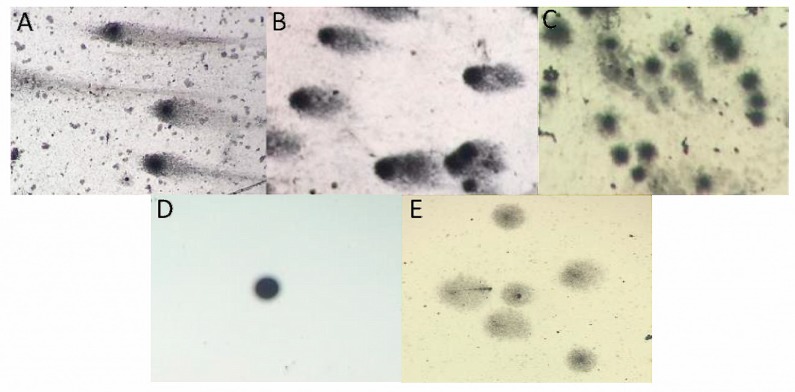
Comet tails of 24 h treatment of lymphocytes with H_2_O_2_ and isolated strains. (**A**) *L. mesenteroides*, (**B**) *L. satsumensis*, (**C**) *S. cerevisiae*, (**D**) negative control (suspension containing only lymphocytes) and (**E**) positive control (suspension containing lymphocytes and hydrogen peroxide).

**Table 1 foods-08-00485-t001:** Antimicrobial activity of *Lactobacillus satsumensis* LPBF1, *Leuconostoc mesenteroides* LPBF2 and *Saccharomyes cerevisiae* LPBF3 against indicator microorganisms.

Microorganism	Inhibition Zone (mm) *	
	*Escherichia coli*	*Staphylococcus aureus*
*Lactobacillus satsumensis* (LBPF1)	12.5 ± 0.50 ^Ca^ **	10.5 ± 0.50 ^Ba^
*Leuconostoc mesenteroides* (LBPF2)	10.5 ± 0.50 ^Ca^	12.0 ± 1.00 ^Ba^
*Saccharomyces cerevisiae* (LBPF3)	8.0 ± 0.10 ^Ca^	8.5 ± 0.50 ^Ba^
Honey kefir beverage	27.5 ± 1.50 ^Aa^	19.5 ± 1.50 ^Ab^
Control (Ampicillin 50 mg/mL)	42.5 ± 1.50 ^Ba^	23.5 ± 0.50 ^Aa^

* Values represent the mean ± standard deviation of three independent experiments. ** Upper-case letters show significant differences between column, and lower-case letters show significant differences between lines, as determined by Tukey´s test (*p* < 0.05).

**Table 2 foods-08-00485-t002:** DNA damage index by H_2_O_2_ after 1 and 24 h of *Lactobacillus satsumensis* LPBF1, *Leuconostoc mesenteroides* LPBF2 and *Saccharomyes cerevisiae* LPBF3 application.

Microorganism	Damage Index *	
	1 h	24 h
*Lactobacillus satsumensis*	3.05 ± 0.25 ^aA^ **	3.45 ± 0.13 ^aA^
*Leuconostoc mesenteroides*	3.26 ± 0.21 ^aA^	3.65 ± 0.24 ^aA^
*Saccharomyces cerevisiae*	2.44 ± 0.06 ^bA^	2.39 ± 0.16 ^bA^
Positive control	3.79 ± 0.19 ^cA^	3.86 ± 0.06 ^aA^

* Values represent the mean ± standard deviation of three independent experiment. ** Means of triplicate in each column bearing the same lower-case letters or the same capital letters in each row are not significantly different (*p* > 0.05) from one another using Tukey’s Test (mean ± standard variation).

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
