# Peer review of "In Vitro Probiotic Properties and DNA Protection Activity of Yeast and Lactic Acid Bacteria Isolated from A Honey-Based Kefir Beverage"

_foods, 2019, doi:10.3390/foods8100485_

Round 1

Reviewer 1 Report

General comments:

The manuscript deals with the determination of the in vitro probiotic properties and DNA protection activity of yeast and lactic acid bacteria isolated from a honey-based kefir beverage.

The manuscript is interesting has an average scientific merit and the experimental methodology seems to be correct. However, there are some issues that should be clarified:

Materials and methods section: Subsection: 2.2.1. Acid tolerance, page 2, lines 76-78: Did the authors use HCl to adjust the pH of the MRS or YM broths (pHi between 6.0 and 6.3) to 7.0? I think that was NaOH. Why did the authors use the broth adjusted to H values 2.0, 3.0, 4.0 and 7.0? To make the acid tolerance assays, the cells should be mixed with PBS buffer (nor the broths) adjusted to the above mentioned pH values. Why did the authors not use buffers solutions?

Figure 1A, 1C, 1D and 1E, page 6. Why did the initial cell counts were different?

Captions for Figure 2, page 7: The heading “3.2. Adhesion ability” should be separated from the figure caption.

Captions for Figure 4, page 9: The name of the different strains should be written in italics.

Author Response

Reviewer #1

General comments:

The manuscript deals with the determination of the in vitro probiotic properties and DNA protection activity of yeast and lactic acid bacteria isolated from a honey-based kefir beverage.

The manuscript is interesting has an average scientific merit and the experimental methodology seems to be correct. However, there are some issues that should be clarified:

Materials and methods section: Subsection: 2.2.1. Acid tolerance, page 2, lines 76-78: Did the authors use HCl to adjust the pH of the MRS or YM broths (pHi between 6.0 and 6.3) to 7.0? I think that was NaOH. Why did the authors use the broth adjusted to H values 2.0, 3.0, 4.0 and 7.0? To make the acid tolerance assays, the cells should be mixed with PBS buffer (nor the broths) adjusted to the above mentioned pH values. Why did the authors not use buffers solutions?

We added the information concerning pH adjustment to 7 with NaOH. Thank you for your attention. Please see lines 76-79.

In addition, to make the acid tolerance assays, the cells were grown in MRS or YM broth. The growth of the microrganisms in these media were evaluated over time.  For this reason, the cells were not only mixed, but cultivated in media with different pHs. These procedures followed those performed by Pienez et al., (2014) Probiotic potential, antimicrobial and antioxidant activities of Enterococcus durans strain LAB18s. Food Control.

 Figure 1A, 1C, 1D and 1E, page 6. Why did the initial cell counts were different?

The initial inoculation was performed by cultures standardized at an absorbance (OD600) = 1.0 ± 0.05. Optimal density does not discriminate viable from nonviable cells, which results in slightly different counts by viable cell plating (pour plate method). Even so, these distinct initial counts did not reflect the resistance dynamics at the measured times.

 Captions for Figure 2, page 7: The heading “3.2. Adhesion ability” should be separated from the figure caption.

Ok, done. Please see line 241.

 Captions for Figure 4, page 9: The name of the different strains should be written in italics.

Ok, done. Please see line 286.

Reviewer 2 Report

In this study, Coelho et al reported in vitro probiotic properties and antioxidant activity of three probiotic bacteria and yeast from Kefir beverage. Research topic seems proper for Foods. This study contain a practical information and should be helpful to readers of this journal. However, I have some concern related to experimental design and they should be revised. I’d like to suggest that authors should address the following questions and concerns.

1) In figure 1, author need to show some another bacterial control, which do not have a strong resistant to acid and bile salts. In antimicrobial activity test, more than log 3 reduction of bacteria, it can be considered as a potent antimicrobial activity. Therefore, authors may emphasize the probiotic properties of their three probiotics as compared to another non-probiotic bacteria.

2) In figure 2, author need to add A and B mark to Figure 2, Why authors did not demonstrate the hydrophobicity of Leuconostoc mesenteroides?

3) In DPPH experiment (figure 5), author just compared the relative antioxidant activity of three probiotics, what is the meaning of this data? it seems not proper. Authors need to compared to some another antioxidant chemicals (ex Trolox) or another bacteria such as Escherichia coli. Authors may indicate the DPPH radical scavenging activity with the Trolox equivalent or relative activity as compared to a standard bacteria such as E. coli.

4) In Figure 6 and Table 2, authors need to clearly describe what is negative control and positive control. The quantitation of DNA damage and their description related to comet images seem not proper. For example, Fig 6A and 6B groups seem to have strong DNA damage than positive control, because their comet tail is long and apparent. I’d like suggest that author may compare the DNA damage between their probiotic bacteria and a control bacteria which do not have preventive activity.

5) Section 3.5, I think authors need to demonstrate their data such as supplementary materials. Did author added some proper positive control such as hemolytic toxin?

1) Scientific name should be italic. For example, line 203 to 208; authors need to revise the entire manuscript.

Author Response

Reviewer #2

In this study, Coelho et al reported in vitro probiotic properties and antioxidant activity of three probiotic bacteria and yeast from Kefir beverage. Research topic seems proper for Foods. This study contain a practical information and should be helpful to readers of this journal. However, I have some concern related to experimental design and they should be revised. I’d like to suggest that authors should address the following questions and concerns.

We appreciate your evaluation and the indication of inaccurate information and error. We have corrected the manuscript for a better understanding and critical information for readers.

1) In figure 1, author need to show some another bacterial control, which do not have a strong resistant to acid and bile salts. In antimicrobial activity test, more than log 3 reduction of bacteria, it can be considered as a potent antimicrobial activity. Therefore, authors may emphasize the probiotic properties of their three probiotics as compared to another non-probiotic bacteria.

We appreciate your observation. However, additional testing will excessively increase the working' time and the publication. The methodology used followed that described in Pienez et al., (2014) Probiotic potential, antimicrobial and antioxidant activities of Enterococcus durans strain LAB18s. Food Control.

 2) In figure 2, author need to add A and B mark to Figure 2, Why authors did not demonstrate the hydrophobicity of Leuconostoc mesenteroides?

The mark “A” in “B” has been added. Please see line 259.

No hydrophobicity ability was observed for Leuc. mesenteroids. This information has also been added to the figure caption (line 260) and text (line 251).

3) In DPPH experiment (figure 5), author just compared the relative antioxidant activity of three probiotics, what is the meaning of this data? it seems not proper. Authors need to compared to some another antioxidant chemicals (ex Trolox) or another bacteria such as Escherichia coli. Authors may indicate the DPPH radical scavenging activity with the Trolox equivalent or relative activity as compared to a standard bacteria such as E. coli.

Methanol was used as a control because it has “zero” oxidizing activity - any value presented above has antioxidant activity. We agreed that the addition of a commercial antioxidant (Trolox) or a standard microbial strain could aid in the quality of the results. We will take this into consideration in our next studies. However, the results with yeast are promising, since it presented higher antioxidant capacity (compared to bacteria in the methodology used), which corroborated the results of DNA protection.

4) In Figure 6 and Table 2, authors need to clearly describe what is negative control and positive control. The quantitation of DNA damage and their description related to comet images seem not proper. For example, Fig 6A and 6B groups seem to have strong DNA damage than positive control, because their comet tail is long and apparent. I’d like suggest that author may compare the DNA damage between their probiotic bacteria and a control bacteria which do not have preventive activity.

It seems that there is a misunderstanding on the results of comet assay. All microbial treatments showed DNA protective action in relation to the control. Despite the occurrence of fragmented DNA tails in the images with the microorganism treatments, it is possible to observe the colloidal structure (or compaction) of the DNA (Fig. 6 a, b, c). On the other hand, in the positive control (Fig. 6 e), there is no preserved DNA structure, which corresponds to that all cells suffered apoptosis. A better explanation of the results has been added. Please see lines 302-306. Negative control and positive control were clarified in the caption of Figure 1. Please see lines 314-315.

5) Section 3.5, I think authors need to demonstrate their data such as supplementary materials. Did author added some proper positive control such as hemolytic toxin?

Positive control was not added following methodology described in Pienez et al., (2014) Probiotic potential, antimicrobial and antioxidant activities of Enterococcus durans strain LAB18s. Food Control. We will take this issue into consideration in future studies.

1) Scientific name should be italic. For example, line 203 to 208; authors need to revise the entire manuscript.

Ok, done. We appreciate your time and attention.

Round 2

Reviewer 2 Report

Now it have revised well.